# The Impact of the COVID-19 Pandemic on Mental Health and Psychological Well-Being of Young People Living in Austria and Turkey: A Multicenter Study

**DOI:** 10.3390/ijerph17239111

**Published:** 2020-12-06

**Authors:** Türkan Akkaya-Kalayci, Oswald D. Kothgassner, Thomas Wenzel, Andreas Goreis, Anthony Chen, Veysi Ceri, Zeliha Özlü-Erkilic

**Affiliations:** 1Department of Child and Adolescent Psychiatry, Outpatient Clinic of Transcultural Psychiatry and Migration Induced Disorders in Childhood and Adolescence, Medical University of Vienna, Währinger Gürtel 18-20, 1090 Vienna, Austria; tuerkan.akkaya-kalayci@meduniwien.ac.at; 2Postgraduate University Program Transcultural Medicine and Diversity Care, Medical University of Vienna, Spitalgasse 23, 1090 Vienna, Austria; 3Department of Child and Adolescent Psychiatry, Medical University of Vienna, Währinger Gürtel 18-20, 1090 Vienna, Austria; oswald.kothgassner@meduniwien.ac.at; 4Department of Psychiatry and Psychotherapy, Medical University of Vienna, Währinger Gürtel 18-20, 1090 Vienna, Austria; thomas.wenzel@meduniwien.ac.at; 5Scientific Section on Psychological Aspects of Torture and Persecution, World Psychiatric Association (WPA), 1226 Thônex, Switzerland; anthonyfchen@gmail.com; 6Department of Clinical and Health Psychology, Faculty of Psychology, University of Vienna, 1010 Vienna, Austria; andreas.goreis@univie.ac.at; 7Outpatient Unit for Research, Teaching and Practice, Faculty of Psychology, University of Vienna, 1010 Vienna, Austria; 8Department of Child Development, Faculty of Health Sciences, Batman University, Batman Üniversitesi, Merkez Kampüsü, 72060 Batman, Turkey; veysiceri@gmail.com

**Keywords:** COVID-19, mental health, well-being, young people, Austria, Turkey

## Abstract

Little is known about the impact of the COVID-19 pandemic on the mental health and psychological well-being of young people. The aim of this study is to investigate the psychological well-being and changes in the mental-health state of young people living in Austria and Turkey. By using an anonymous online survey, we recruited 1240 people aged 15–25 years from these two countries. We used the “Psychological General Well-being” and a self-created questionnaire to capture individual experiences during the COVID-19 pandemic and quarantine period. The native Turks indicated higher “anxiety” (*p*s < 0.010), lower “vitality” (*p*s < 0.011), and lower “general health” (*p*s < 0.011) than native Austrians or Austrian migrants and increased “depression” (*p* = 0.005) and lower “self-control” (*p* = 0.022), than Austrian migrants. Moreover, 50.9% of native Turks reported a decrease in their mental health status, compared to 31.1% of native Austrians and 23.7% of Austrian migrants. Participants with financial problems (OR = 1.68) and prior mental health problems (i.e., already in treatment by the time of COVID-19, OR = 5.83) reported a higher probability for a worsening in their mental health status. Our results show that the COVID-19 pandemic impaired the psychological well-being and mental health of young people. Especially people in Turkey were most affected, probably due to the stringent policies to fight COVID-19.

## 1. Introduction

In early 2020 the World Health Organization (WHO) confirmed the outbreak of a new coronavirus disease (COVID-19), and by March 2020, COVID-19 was confirmed to be a pandemic [1]. Currently, COVID-19 has become a problematic global threat as it has spread to around 210 countries [2].

Austria as well as Turkey adopted lockdown strategies to contain COVID-19, which had different restrictions for public, professional, and private life. Examples include the closure of schools and businesses and the enforcement of social distancing. People who did not obey the limitations had legal penalties, mostly fines, and the legal base of such measures have been challenged in Austria even up to the constitutional court. Because these measures are in the case of a national health emergency but potentially place restrictions on human rights, it is especially important to examine the scientific basis of such justifications and their possible impact on public health, human rights, and the medico-legal system.

The rate of COVID-19 infection and mortality among children and adolescents has been relatively low. Nevertheless, an important concern is how to deal with lockdown and quarantine measures, which may create distress and consequently push sensitive young people into personal crises and destabilize families as protective frameworks for these situations [3,4].

### 1.1. Impact of COVID-19 Pandemic on Well-Being and Psychological State

The outbreak of COVID-19 and measures attempting to deal with the pandemic have had a strong impact on the well-being [5] and mental health [6] of individuals. The rapid transmission and death caused by COVID-19, the limited knowledge of the threat, lack of effective treatments, and strict quarantine restrictions such as suspension of school and physical social networks have led to diverse mental health problems like fear, anxiety, depression, and sleep problems throughout the population [7,8]. The psychological reactions of children and young people during a pandemic are quite similar to those of adults, but the age, level of understanding, and innate coping skills of young people play a crucial role [9,10,11]. Moreover, the impact of the pandemic on children and adolescents who already have psychological problems compared to those who do not may vary [12]. For example, first published data indicates that among youths living in the Ukraine, suicide attempts, severe self-harming behavior, and psychotic disorders have increased during the COVID-19 pandemic [12].

As social contact plays an important role in children’s normal psychological development and well-being [11], restrictions of social life during the COVID-19 pandemic may particularly impair young people’s well-being [2]. School closures and social distancing may be especially difficult for adolescents, as during adolescence, they grow in independence, and connections to peers have high importance. Moreover, separation from caregivers and friends may increase the risk of psychiatric disorders among young people. Previous studies show that children who were isolated or quarantined during pandemic diseases developed acute stress disorder, adjustment disorders, and post-traumatic stress disorder [13]. Furthermore, prolonged school closure and home confinement may lead to an increase in anxiety, frustration, boredom, depression, and posttraumatic stress disorder symptoms among young people [6,9]. School closures bear some further risks, as documented during the Ebola epidemic in West Africa, when the rates of child labor, neglect, sexual abuse, and adolescent pregnancies peaked. During the lockdown, more than 1.5 billion young people, over 90% of the world’s students, were out of school [2].

Psychiatric patients are also physically a more vulnerable population to COVID-19 disease [14], as they generally have a lower life expectancy and poorer physical health outcomes compared to the general population [15]. For instance, in China, a great number of patients with psychiatric disorders were infected [7]. Furthermore, in many countries including Austria and Turkey, during the lockdown, access to mental-health treatment was usually restricted to only emergency treatments [16].

### 1.2. Migration-Related Problems during the COVID-19 Pandemic

Migrants living in Austria have lower income levels, translating into poorer living-conditions and lower socio-economic status compared to the native population [17]. These facts may lead to lower access to necessary home-schooling equipment and parental support among migrants. Intra-familial conflicts are also more common [18]. Poor housing conditions, as well as less individual learning and private space during the curfews due to the COVID-19 pandemic, might exacerbate these conflicts among migrant families compared to non-migrant ones.

We assume that even though the COVID-19 pandemic will affect all young people, it will affect migrant populations stronger and in different ways, reflecting various socio-economic and psychosocial differences. We further postulate that the COVID-19 pandemic environment will particularly impair the psychological well-being of the migrant population due to the above-mentioned disadvantaged circumstances, which might require special targeted support measures. We, therefore, intend to identify differences in the psychological reaction patterns and needs between migrant and non-migrant young people, as this type of study has not yet been done but is urgently needed to guide measures and interventions in at least Turkey and Austria.

### 1.3. Cross-National Comparison of the COVID-19 Pandemic

During the COVID-19 pandemic, Turkey has implemented more stringent domestic quarantine policies than Austria. For example, people under 20 years of age were required to comply with a mandatory 24-h curfew for many weeks. Likewise, in some instances, there was a total ban on leaving homes for the whole population. In contrast, people living in Austria did not face such strict limitations during the COVID-19 pandemic, as an absolute curfew was not declared for any part of the population, even during the main lockdown period, leading to a different psychosocial environment than Turkey. Therefore, we will compare the general psychological well-being, attitude concerning the COVID-19 pandemic, and impact of the COVID-19 pandemic among different populations in these two countries.

To our best knowledge, there are so far no published studies conducted in Austria that explicitly compare the impact of the COVID-19 pandemic on psychological well-being between Austrian (non-migrant) and migrant youths and young adults aged between 15–25 years, nor is there any transcultural comparison on this question between countries with different lockdown strategies.

Further, there is in general very limited research on how pandemic outbreaks impact psychological well-being [19,20], or on factors which increase risk or improve resilience in such situations [5]. Therefore, the aim of the study was to analyze the transcultural differences between migrant and non-migrant youths and young adults aged between 15–25 years living in Austria and their psychological well-being during the COVID-19 pandemic.

We postulated that the psychological well-being of young people with a migration background in Austria will be more negatively affected by the pandemic in comparison to their native peers. Moreover, we hypothesized that the COVID-19 pandemic negatively affects the psychological well-being of non-migrant young people in Turkey more than that of their peers in Austria.

Our first hypothesis was that psychological well-being during the COVID-19 pandemic would be lower among the study subjects with migration a background than in their non-migrant peers. Furthermore, we assumed that in the PGWB subscales, participants with a migration background would have significantly higher scores in “anxiety” and “depressed mood” compared to non-migrant peers. We further expected significant general psychological differences between young adults living in different lockdown situations in Austria and Turkey, given different stressors and limitations as noted above.

## 2. Materials and Methods

We used an anonymous online survey for data collection created by the “SoSci Survey” program. The study subjects were recruited via different social media channels, which are widely used by the target age groups, such as Facebook, Instagram, WhatsApp, etc., with the goal of generating a representative sample of the young population. Due to the aim of the current research the inclusion criteria for both Austria and Turkey, sites were an age between 15–25 years, living in the respective country, and sufficient language skills in the respective country’s language.

We used the validated German [21] and Turkish [22] versions of the “Psychological General Well-being” index (PGWB) of DuPuy et al. 1984 [23], which consists of 22 items on 6-point Likert scales, divided into 6 subscales: Anxiety, Depressed mood, Positive well-being, Self-control, General health, and Vitality. The PGWB captures the general well-being during the last month, but for the present study, we extended the query to refer to the last two months to cover quarantine time at the outset of the pandemic. Importantly, and in line with the manual, all subscales of the PGWB are scored such that higher values indicate greater well-being (i.e., a higher score in the anxiety and depression subscales mean fewer indications of anxiety and depression). The psychological impact of the COVID-19 pandemic was measured using the PGWB, and a questionnaire we developed for the study to assess socio-demographic data and individual experiences and changes during the COVID-19 pandemic and quarantine period. These items included prior psychological treatment because of pre-existing mental health problems, deterioration of mental health because of COVID-19 pandemic (response options: 1 = improved, 2 = deteriorated, 3 = unchanged), a proxy item for socioeconomic status (financial problems because of COVID-19 pandemic (1 = yes, 0 = no)), ruminations about COVID-19, fear of being infected, fear relating to infection of a family member, belief in exaggerated measures, and estimated severity of COVID-19 risks (all on 5-point Likert scales). The data were collected from 22 May to 19 June 2020. All study participants provided electronic informed consent before starting the online survey. The structured online survey took approximately 10 min to complete, while only data with a complete set of responses were included.

The study was conducted in accordance with the Declaration of Helsinki, and the protocol was approved by the Ethics Commission of the Medical University of Vienna (protocol number: EK 1488/2020).

### Statistical Data-Analysis

Data were analyzed using IBM SPSS v23 (IBM Corporation, Armonk, NY, USA) [24], with a significance level of α < 0.05. Descriptive differences in proportions between groups were analyzed using χ^2^-tests. For our hypotheses, we conducted three-way MANVOAs to analyze differences between the three groups (Austrian natives, migrants in Austria, and Turkish natives), while accounting for the effect of financial problems (yes/no) and current treatment (yes/no). The outcome variables were (a) the scales of the questionnaires pertaining to COVID-19 and (b) the subscales of the PGWB. Main effects and interactions are reported, and pairwise comparisons (Bonferroni), as well as simple effects analyses (where appropriate), were conducted to break down interactions and control for multiple testing. Age and gender were entered as covariates and controlled for in all analyses. To analyze the effects and differences on the potential deterioration of mental health, we conducted a binary logistic regression. The outcome was the variable “decrease in mental health” and the reference category was a culmination of the two response options “my mental health increased” and “my mental health stayed the same”. Odds Ratios (OR) are reported for the predictors group, financial problems, and current treatment in this analysis.

## 3. Results

### 3.1. Participants

We used an anonymous online survey to collect data for the present study. We analyzed the data from *N* = 1240 participants in total, from two countries (Austria and Turkey). As the participants were only able to progress in the survey after completing previous sections, there was no missing data in the included questionnaires. The exclusion criteria were living outside of Austria or Turkey, having no sufficient German or Turkish language skills, and being outside the age range of 15–25 years.

For 262 of the 1115 total received responses from the Austrian sample and 255 out of 642 responses from the Turkish sample, the responses were excluded from the study due to incompleteness of the questionnaires or due to the defined age range.

In the native Austrian sample (*n* = 621) 70.2% were female, 29.0% were male and 0.8% reported another gender with a mean age of 20.24 (*SD* = 1.622). The sample of migrants in Austria (*n* = 232) consisted of 56.9% female, 42.7% male, and 0.4% participants marking “other” as gender with a mean age of 19.22 (*SD* = 1.628). The participants from Turkey (*n* = 387) consisted of 26.4% female and 73.4% male participants with 0.3% gender given as “other”. The mean age in this sample was 21.13 (*SD* = 1.062). The three groups were different in terms of gender (χ^2^(4, *N* = 1240) = 190.628, *p* < 0.001) and age (F(2, 1273) = 24.568, *p* < 0.001).

We found that 8.9% of the native Austrian sample (*n* = 55), 8.6% of the migrants in Austria (*n* = 20), and 25.6% of young Turks living in Turkey (n = 99) were already in psychological treatment due to pre-existing mental health problems. This difference in proportions was significant (χ^2^(2, *N* = 1240) = 62.212, *p* < 0.001). In terms of adversity (i.e., our financial problems item), 12.2% of native Austrians (*n* = 76) indicated financial problems because of COVID-19 pandemic, compared to 19.4% of migrants in Austria (*n* = 45) and 38.8% of Turks living in Turkey (*n* = 150). This comparison was also significant (χ^2^(2, *N* = 1240) = 99.204, *p* < 0.001).

#### Differences between Different Groups Regarding COVID-19 Related Cognitions

Linear regression analyses showed that identifying as female was associated with a higher fear of being infected, a higher fear of infections of family members, and more rumination about COVID-19 (interactions age × gender, *p*s < 0.005). The interactions further indicated that these associations became weaker with increasing age, being more pronounced in younger individuals (as coefficients were negative, *b*s = −0.06–0.10). Age was positively associated with higher estimated severity of COVID-19 (main effect *p* < 0.001, no significant (sig.) interaction). Furthermore, being younger was associated with higher beliefs that measures against COVID-19 were exaggerated (*p* = 0.001).

As depicted in Figure 1, MANCOVA indicates that people living in Austria (including Austrian natives and migrants living in Austria) reported lower estimates of the severity of COVID-19 infection compared to native Turks (F(2, 1226) = 44.491, *p* < 0.001). Pairwise comparisons revealed that there was no difference in this aspect between native Austrians and migrants (*p* = 0.433) but a significant difference between them and the (non-migrant) group of Turks living in their home country (*p*s < 0.001).

Regarding the fear of being infected with COVID-19 (F(2, 1226) = 81.236, *p* < 0.001) and fear that a family member might be in future infected with COVID-19 (F(2, 1226) = 34.695, *p* < 0.001), post hoc analyses showed that native Austrians reported fear of being infected by COVID-19 to a lower degree than migrants living in the same country (*p* = 0.022) and also lower than Turks living in Turkey (*p* < 0.001). Furthermore, migrants living in Austria reported lower levels of fear of being infected by COVID-19 than Turks living in Turkey (*p* < 0.001). For the fear of the risk of infection of a family member, native Turks living in Turkey reported the highest scores, while migrants in Austria reported lower scores, and native Austrians reported the lowest scores, with significant group differences (*p*s < 0.001 and 0.004). Regarding the question of whether the measures against the COVID-19 outbreak were exaggerated, there was also a significant main effect between the groups (F(2, 1226) = 15.826, *p* < 0.001). Migrants living in Austria and native Austrians reported the highest scores (which did not differ between them, *p* > 0.90)—yet both groups differed from Turks living in Turkey, who reported significantly lower scores (*p*s < 0.001). Moreover, native Turks think significantly more often about COVID-19 (i.e., the rumination item) than native Austrians and their local migrant peers (F(2, 1226) = 12.829, *p* < 0.001; post hoc tests *p*s < 0.001 and 0.006).

None of the interaction terms of influential factors (i.e., group × financial problems, group × treatment, or the three-way variant of it) were significant in any of our five outcomes.

### 3.2. Effects of COVID-19 Pandemic on Psychological Well-Being

Linear regression revealed that female gender was associated with worse outcomes in the subscales of anxiety, depression, self-control, and general health, with these associations more pronounced in younger individuals and tending to wear off with increasing age (interactions *b*s = 0.89–1.4, *p*s = 0.004–0.019); higher scores reflect higher well-being across subscales. Neither gender, age, nor the interaction was significant predictors in the subscale personal well-being or vitality (*p*s > 0.206).

The three groups of participants were compared directly with regard to their psychological well-being. Results are depicted in Figure 2. Results of the MANCOVA revealed significant differences between groups in the subscales of anxiety (F(2, 1226) = 17.559, *p* < 0.001), depression (F(2, 1226) = 5.535, *p* = 0.004), self-concept (F(2, 1226) = 3.812, *p* = 0.022), vitality (F(2, 1226) = 7.597, *p* = 0.001), and general health (F(2, 1226) = 14.473, *p* < 0.001) but not in positive well-being (F(2, 1226) = 1.433, *p* = 0.239). In detail, post hoc analyses showed that for anxiety, native Turks reported more anxiety, followed by native Austrians, then migrants in Austria (all comparisons: *p*s < 0.011). In the depression subscale, only native Turks reported higher values than Austrian migrants (*p* = 0.005). A similar pattern of results was in the subscale self-control—only native Turks differed from Austrian migrants (*p* = 0.019), with native Turks reporting a lower level of self-control. Regarding the subscales of vitality and general health, both native Austrians and their migrant peers reported significantly better scores than native Turks (*p*s = 0.015 and 0.002 in vitality; *p*s < 0.001 in general health).

The effects of our socioeconomic proxy item (i.e., financial problems) and currently being in treatment are shown—and sorted by group—in Table 1. Financial problems were associated with worse psychological well-being in most subscales, regardless of the investigated group. Only the group of native Turks with and without financial problems, however, did not differ in the subscales of anxiety (*p* = 0.214), self-control (*p* = 0.341), positive well-being (*p* = 0.736), and general health (*p* = 0.424). Current treatment—in comparison to no current treatment—was, on average, associated with better scores across all subscales of psychological well-being, with no differences between any of the three groups (*p*s < 0.028).

### 3.3. COVID-19 Pandemic and Mental Health

Of native Turks, 50.9% reported a decrease in their mental health status, whereas 31.1% of native Austrians and 27.7% of migrants in Austria stated this in our study. This difference was significant (χ^2^(2, *N* = 1240) = 59.132, *p* < 0.001. Logistic regression showed that—compared to native Turks—native Austrians (OR: 0.50, *p* < 0.001, 95% CI 0.35–0.73) and Austrian migrants (OR: 0.31, *p* < 0.001, 95% CI 0.19–0.51) had lower odds for a decreased mental health status. Participants with financial problems had a higher probability for a decreased mental health status (OR: 1.68, *p* = 0.022, 95% CI 1.08–2.62). Furthermore, those already in treatment had a higher probability to report a decrease in mental health status (OR: 5.83, *p* < 0.001, 95% CI 3.36–10.13). The effect of financial problems and treatment were similar across all three groups as interactions (i.e., group × financial problems; group × treatment) and were all non-significant (*p*s > 0.054).

## 4. Discussion

The results of the present study show that, in general, both the global COVID-19 pandemic and associated events such as the lockdown strategies might have an adverse impact on the psychological well-being and mental-health state of young people aged 15–25 years. Similarly, Faviere et al. 2020 [25] showed that in the Italian population lower rates of psychological well-being were found just weeks after the outbreak of the COVID-19 pandemic. Similar to our data, the study of Favieri et al. 2020 [25] also reported higher anxiety and depressive symptoms, negative well-being, perception of loss control, less vitality, and lower general health during the COVID-19 pandemic in the Italian population [25]. This was also observed in our study. However, the adverse impact was highest in native Turks living in their home country. As financial problems were associated with worse psychological well-being in most subscales in all investigated groups, there was no difference between native Turks with and without financial problems in the subscales of anxiety, self-control, positive well-being, and general health. This might be because of stringent lockdown in Turkey having a more adverse impact on the psychological well-being of young people living there than in Austria. Similarly, native Turks reported significantly more of a decrease in their mental health status than native Austrians and migrants in Austria, although, in all groups, participants with financial problems and those already in treatment had a higher probability to report a decrease in mental health status. Financial problems must be expected to increase due to both pandemics and pandemic control measures like lockdowns and can themselves induce mental health problems or worsen the state of those having pre-existing mental health problems [26]. Furthermore, it is possible that the larger COVID-19 case number, the stringent lockdown during Turkey’s outbreak, different mental health care service capacity and acceptance, or general socioeconomic conditions that interact with the impact of the pandemic might explain or at least contribute to the adverse impact on the mental health of young people, especially in Turkey.

Additionally, females and younger participants reported not only higher fear of being infected, higher fear of infection of family members, and more rumination about COVID-19, but they also had more anxiety and depression and worse self-control and general health. This means that strict quarantine measures to overcome the COVID-19 pandemic can also potentially, as noted previously, result in negative mental health outcomes, particularly in vulnerable youths such as those who already had earlier mental health problems [10].

Increased support and low-barrier treatment offers, potentially including online services [27], should therefore be actively offered to those with previous or ongoing treatment for mental health problems, even when services are increasingly focusing on COVID-19 infected patients. This should, in the context of our data, be especially considered for females and younger populations that suffer, as noted previously, from already disadvantaged financial situations [28]. Therefore, mental health services and social support for young persons during pandemics should receive more attention. Future research should explore the interaction of socioeconomic stress, existing mental health problems, and policies to address this challenge, in these countries as well as others, including different ethnic groups in developing countries, and those with already less stable healthcare systems during the pandemic. Further research also needs to focus on the extent of the measures and their impact on mental state. It may be that there are factors beyond a simple lockdown that can contribute to the answer, as noted by the fact that Turkey implemented stringent polices to combat the COVID-19 pandemic, but the number of new infections during and after the lockdown was higher than Austria’s, which did not implement the same severe restrictions. Furthermore, our data indicate that psychological problems may be due to the strict measures and increased isolation in a group-oriented society, and this should be considered as dangerous collateral damage in this context. This should be kept in mind during the planning of restrictions, especially for the Turkish health authorities.

Fear of infection and the expectation of a COVID-19 infection as being a serious health risk were significantly higher in Turkey compared to migrant and non-migrant Austrian groups, in this study. Furthermore, regarding the fear of infection of a family member, Turks stated the highest fear scores, while native Austrians had the lowest scores. The results of this study also show that young people with a migration background—at least in Austria—have more fear than natives that their family members could get COVID-19, which might reflect a more group-oriented (“universalistic”) cultural background focusing more on the family than on the individual. Health belief models, usually shaped by the cultural and sociopolitical environment, including migration status and acculturation, should also be considered in this context. Health-related beliefs can be seen as causing psychological distress but could also be seen as motivating populations towards careful behavior and compliance or non-compliance with COVID-19 related public health campaigns. Culture-specific health belief models should therefore be explored in further research to better address the respective concerns in different ethnic and social groups. Clinicians active in mental health services should also take these factors into consideration and, if possible, offer more time to the social, economic, and cultural risk groups described in our study to address the above factors.

## 5. Conclusions

Public health restrictions are necessary to overcome the COVID-19 pandemic to avoid illnesses and mortalities, but it should be noted that they also contribute to negative psychological outcomes for the population. We conclude that under conditions of a pandemic and measures to respond to a pandemic, youths and young adults may be particularly vulnerable to mental health problems, especially those with pre-existing psychological problems. Our data does not address the question of specific mechanisms in how the restriction of younger populations affects the spread of COVID-19 disease, though this has been looked at in other studies. Regardless, the impact of the pandemic’s presence and the strict social measures adopted by countries must be examined as a potential adverse event for vulnerable youth. Policies that consider the importance of mental health and well-being of individuals and also vulnerable age or social groups during a pandemic have not been adequately studied by existing research [29]. Additional supportive measures might be specifically required for females, younger persons, and migrants, as our data indicates that the influence of the general social environment and utilized lockdown strategies might be important factors to be considered in such a crisis. Nevertheless, developing “diversity-care” measures for youths and young adults with and without migration background might require reorganization. Further, we believe that research and intervention strategies should also explore the situation of migrants and refugees in Tukey and other countries with severely challenged economic and health care situations. Our findings may provide vital guidance for developing culture-, language-, gender-, and age-sensitive measures for adequate treatment of young people with and without migration background in similar pandemics or other natural disasters and indicate that an understanding of both culture and the socioeconomic factors including financial background are required to identify risk groups in need of greater access to specialized mental health services. Finally, it should be considered that Turkey itself, as also other countries in the region, houses a large number of migrants with diverse background cultures under adverse conditions. This includes internal migrants moving from traditional rural communities to cities and migrants and refugees from neighboring war zones such as Syria [30]. They suffer from the impact of displacement, marginalization, and financial and other hardships, even more than migrants in Austria, and have not yet been covered by our or other’s research. This issue should therefore also be addressed urgently, as highlighted in our data on the complex interaction between culture and socioeconomic factors influencing mental health during a pandemic.

While strict lockdown measures are necessary for the success of overcoming the COVID-19 pandemic, we must also consider their impact on the mental health status of individuals and identify the allocation of resources to combat this collateral damage.

### Limitations

As there was no equal distribution of the genders, there is a sampling bias for gender. The majority of the study participants in Austria were females (native Austrians: 70.2%, migrants in Austria: 56.9%), while in Turkey males (73.4%) exceeded females.

As we have no baseline differences analysis between the countries before the outbreak of the pandemic, the results of our study may be influenced by pre-existing cultural differences.

Online surveys mostly bear biases such as coverage-bias, which influences the representativeness of the study sample, and social desirability bias, which can be common due to self-reporting.

To our best knowledge, there is no study explicitly comparing the psychological outcomes of the COVID-19 pandemic in Austria and Turkey. Our study contains new scientific knowledge, and therefore, it should be considered as the first step in this research. Further studies are needed to examine this topic.

## Figures and Tables

**Figure 1 ijerph-17-09111-f001:**
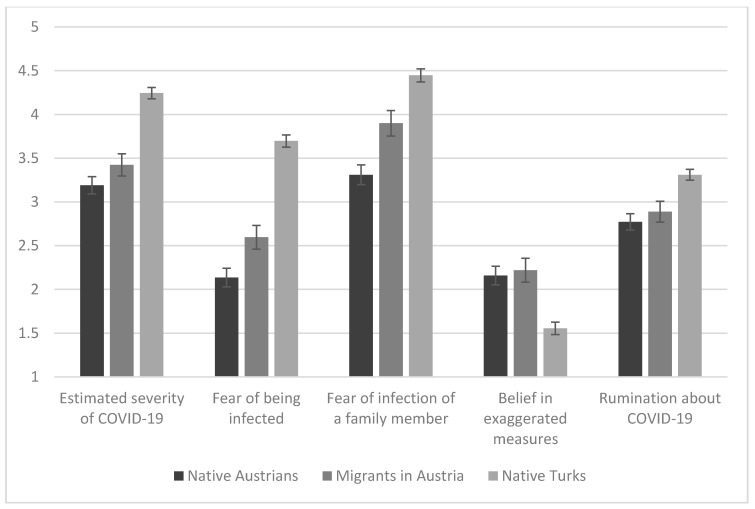
COVID-19 related cognitions with *M* and *SEM* in the different groups. The means are depicted after controlling for the effect of age and gender. The y-axis indicates the possible range of the ratings of the items.

**Figure 2 ijerph-17-09111-f002:**
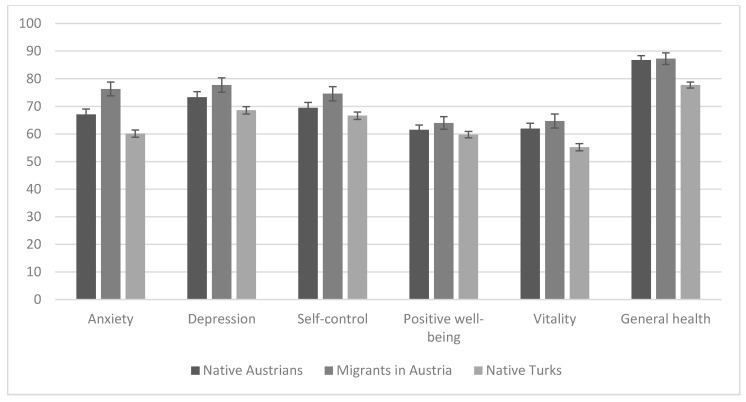
Subscales of the psychological general well-being index with *M* and *SEM* in the different groups. The means are depicted after controlling for the effect of age and gender. Higher scores indicate more well-being (i.e., lower levels of anxiety but higher levels of vitality). The y-axis indicates the possible values of the scores of the subscales.

**Table 1 ijerph-17-09111-t001:** MANCOVA results regarding group comparisons of the psychological general well-being index separately depicted with the factors financial problems and current treatment.

Outcome	Group	Financial Problems	*M* (*SEM*)	F (Financial Problems × Group)	Current Treatment	*M* (*SEM*)	F (Current Treatment × Group)
Anxiety	Native AT	**Yes (*n =* 76)**	61.25 (3.49)	**4.211 ***	**Yes (*n =* 55)**	49.38 (3.6)	**4.590 ***
		**No (*n =* 545)**	72.94 (1.62)		**No (*n =* 566)**	84.81 (1.36)	
	Migrant AT	**Yes (*n =* 45)**	67.67 (3.82)		**Yes (*n =* 20)**	67.44 (4.61)	
		**No (*n =* 187)**	84.89 (3.19)		**No (*n =* 212)**	85.12 (1.89)	
	Native TR	**Yes (*n =* 150)**	58.62 (1.83)		**Yes (*n =* 99)**	47.84 (2.13)	
		**No (*n =* 237)**	61.63 (1.69)		**No (*n =* 288)**	72.41 (1.31)	
Depression	Native AT	**Yes (*n =* 76)**	68.18 (3.63)	1.303	**Yes (*n =* 55)**	53.41 (3.74)	**5.749 ***
		**No (*n =* 545)**	78.43 (1.68)		**No (*n =* 566)**	93.2 (1.41)	
	Migrant AT	**Yes (*n =* 45)**	70.13 (3.97)		**Yes (*n =* 20)**	66.79 (4.79)	
		**No (*n =* 187)**	85.32 (3.32)		**No (*n =* 212)**	88.65 (1.96)	
	Native TR	**Yes (*n =* 150)**	65.4 (1.9)		**Yes (*n =* 99)**	56.03 (2.21)	
		**No (*n =* 237)**	71.73 (1.76)		**No (*n =* 288)**	81.1 (1.36)	
Self-control	Native AT	**Yes (*n =* 76)**	65.6 (3.63)	1.368	**Yes (*n =* 55)**	50.5 (3.74)	**3.244 ***
		**No (*n =* 545)**	73.27 (1.68)		**No (*n =* 566)**	88.37 (1.42)	
	Migrant AT	**Yes (*n =* 45)**	69.21 (3.98)		**Yes (*n =* 20)**	63.81 (4.8)	
		**No (*n =* 187)**	79.94 (3.32)		**No (*n =* 212)**	85.34 (1.96)	
	Native TR	**Yes (*n =* 150)**	65.4 (1.91)		**Yes (*n =* 99)**	51.69 (2.21)	
		**No (*n =* 237)**	67.8 (1.76)		**No (*n =* 288)**	81.51 (1.37)	
Positive well-being	Native AT	**Yes (*n =* 76)**	58.54 (3.19)	2.670	**Yes (*n =* 55)**	50.46 (3.28)	**3.164 ****
		**No (*n =* 545)**	64.41 (1.48)		**No (*n =* 566)**	72.49 (1.24)	
	Migrant AT	**Yes (*n =* 45)**	58.08 (3.49)		**Yes (*n =* 20)**	58.65 (4.21)	
		**No (*n =* 187)**	69.87 (2.91)		**No (*n =* 212)**	69.31 (1.72)	
	Native TR	**Yes (*n =* 150)**	59.36 (1.67)		**Yes (*n =* 99)**	53.58 (1.94)	
		**No (*n =* 237)**	60.11 (1.55)		**No (*n =* 288)**	65.89 (1.2)	
Vitality	Native AT	**Yes (*n =* 76)**	57.99 (3.51)	0.947	**Yes (*n =* 55)**	50.37 (3.61)	2.041
		**No (*n =* 545)**	65.85 (1.62)		**No (*n =* 566)**	73.47 (1.37)	
	Migrant AT	**Yes (*n =* 45)**	57.8 (3.84)		**Yes (*n =* 20)**	59.17 (4.63)	
		**No (*n =* 187)**	71.56 (3.21)		**No (*n =* 212)**	70.19 (1.9)	
	Native TR	**Yes (*n =* 150)**	52.16 (1.84)		**Yes (*n =* 99)**	47.17 (2.14)	
		**No (*n =* 237)**	58.27 (1.7)		**No (*n =* 288)**	63.26 (1.32)	
General Health	Native AT	**Yes (*n =* 76)**	82.58 (2.95)	1.627	**Yes (*n =* 55)**	75.86 (3.04)	1.160
		**No (*n =* 545)**	90.89 (1.37)		**No (*n =* 566)**	97.6 (1.15)	
	Migrant AT	**Yes (*n =* 45)**	84.43 (3.23)		**Yes (*n =* 20)**	80.4 (3.9)	
		**No (*n =* 187)**	90.12 (2.7)		**No (*n =* 212)**	94.15 (1.6)	
	Native TR	**Yes (*n =* 150)**	76.87 (1.55)		**Yes (*n =* 99)**	67.89 (1.8)	
		**No (*n =* 237)**	78.51 (1.43)		**No (*n =* 288)**	87.49 (1.11)	

Note: significant results are in bold; * *p* < 0.05, ** *p* < 0.01.

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
