# Peer review of "The Impact of the COVID-19 Pandemic on Mental Health and Psychological Well-Being of Young People Living in Austria and Turkey: A Multicenter Study"

_ijerph, 2020, doi:10.3390/ijerph17239111_

Round 1

Reviewer 1 Report

This study of mental health concerns in two countries provides insight into both the potential impact of COVID for youth and young adults as well as between those who may be most vulnerable to mental health struggles (migrants and those with pre-existing concerns). The authors provide sufficient information on combined mean scores to allow other researchers to build on these findings. The discussion section to a large degree restates the results and should be rethought. The authors need to include a paragraph on the limits of the study and to suggest future research. The authors may have thoughts that would inform clinicians and policy including the importance of access to mental health services in times of crisis. The conclusion section does include notes that are better addressed in the discussion and then underlined in the conclusions. Specific suggestions and edits follow.

  1. 2 line 75 quarantined vs. guaranteed?

line 91 situation should be plural

  1. 3, lines 118 and 119, “in general” occurs twice in same sentence, please cut one
  2. 3, line 145 please clarify how the variable “belief in exaggerated measures” was phrased in the survey.

The authors should better describe consent process and if there were methods in the survey to ensure the target audience, especially for the age range, were included.

  1. 4, line 149, The authors need to say how many total responses were received and the number of cases excluded due to incompleteness. To the extent that they can, the authors need to discuss the sample representativeness. Especially for gender, there is sampling bias. The authors in the discussion need to note this as a limit to the study and talk about how this may have affected the results. The authors may want to include a few sentences that report sample differences by country/migrant status for gender and age.

Lines 164-66 should go after line 149, not in statistics section.

Line 180 clarify “them” – is this a combined migrant and native Austrian comparison or 2 separate tests?

3.1.1 is a little confusing. Perhaps separate findings for Austrian migrant and native comparison into 1 paragraph and the country comparisons is a second paragraph.

  1. 6 Please correct the formatting in lines 223-4

Table 2 compares respondents who were in- versus not in treatment. In the results text or table 2, the authors should note sex/age differences in MH/COVID measures.

  1. 8 lines 266 to 268. Although the measures to contain COVID 19 are certainly a big difference between the two comparison countries, this is not the only significant difference. The authors need to be more careful in attributing the differences solely to pandemic response. The authors should note other cultural differences that may also account for the differences. Specifically, because there is no baseline difference analysis between the countries, there could have already been the same difference pre-existing and this should be noted in the limitations.

The authors need to move the first two sentences (lines 265 to 269) out of the current paragraph as the main content in the paragraph has to do with the important finding about those who already had pre-existing mental health problems. The speculation takes away from this content. Perhaps this is better put in the paragraph on p. 9, lines 304-306. The authors can certainly mention this possibility but need to acknowledge it as speculation and perhaps call for research that could better identify restrictions – pandemic response or not – and the link to mental health.

The authors should also go through the discussion section and take out redundancy with the results. This section to a large degree restates the results. Instead, the authors need to strengthen the links to further research and also suggest where future research needs to focus.

The discussion section would also be strengthened if the authors talked about how these mental health concerns need to be addressed clinically. For those who are already in treatment or who have pre-existing mental health concerns, what do clinicians need to look at? Under lockdown conditions, how may that affect treatment?

For policy considerations, access to care for migrants may be difficult in both countries. Under conditions of a pandemic, what needs to be done?

The discussion section is missing a section on the limits of the study. The authors need to describe the potential bias of doing online research, the lopsided gender distribution for the different samples, bias due to social desirability, and others. Then end with an affirmation of the importance of their study as a first step in this research.

The first sentence of the conclusion is unjustified as noted above (lines 308-310). This is pure speculation. The study design does not afford this conclusion. Perhaps a better approach may be to say that under conditions of a pandemic and measures to respond to a pandemic, youth and young adults may be particularly vulnerable to mental health problems, especially if pre-existing concerns are present.  These findings are based on self-report which may or may not reflect “true” worsening of conditions.

The conclusions do note some of the concerns noted above as relevant for the discussion section. The conclusions will be stronger if they build on many of the issues in the discussion section.

Reviewer 2 Report

Overall, the document is well written and has been enjoyable to read. The authors aimed to determine the Impact of the COVID-19 pandemic on the well-being and psychological state of young people, taking into account the problems related to migration, previous psychological treatment and other variables.

The manuscript needs some changes to improve it.

The hypotheses should be grouped at the end of the introduction section. Change “Our first hypothesis was that the psychological well-being during the COVID-19 pandemic 150 would be lower among the study subjects with migration background then in their non-migrant 151 peers. Furthermore, we assumed that in the PGWB subscales, participants with a migration 152 background would have significantly higher scores in “anxiety” and “depressed mood” compared 153 to non-migrant peers. We further expected significant general psychological differences between 154 young adults living in different lockdown situations in Austria and Turkey, given different stressors 155 and limitations as noted above.” from the material and methods section. Thios section should include the hypotheses related to differences between those in psychological treatment and those not.

Statistical section is incomplete: At line 236: risk (RR: 1.29, 95% CI 1.068-1.555) have been obtained by linear regression? Then include those analyses at material and methods section. At line 226 some percentages are showed but chi-square data is not included neither in the Statistical section.

Financial problems have been used to evaluate its impact on the psychological well-being, what about other variables? Sex, age, etc. If some are relevant try a multivariate linear regression. The relationship between individual experiences and the Psychological Well-being should be evaluated. If some significative correlations or linear regressions appear, try to build a structural equation.

The conclusions about financial problems, and other variables related, have been omitted.

Few minor changes suggested:

Line 224: “and 223 those not,”

Table 2, anxiety 18.766 in Migrants in Austria, negative?

Reviewer 3 Report

The work entitled “The impact of the COVID-19 pandemic on mental 3 health and psychological well-being of young people living in Austria and Turkey: a multicenter study” contains new scientific knowledge and covers a relevant topic. However, I have some comments that have to be addressed before it can be considered for publication.

The participants sections should provide information about tha nature of the sampling (e.g. incidental, stratified, etc), and how authors handle with missing values if any. In addition exclusión criteria if any should be mentioned.

Also, did authors check for previous history of mental health problems. It seems pretty likely that out of 1240 participants some of them have or have had mental health problems.

The statistical analysis section should provide information about the variables analyzed and the rationale for that. Also, why authors decided to conduct an ANOVA instead a MANOVA or a MANCOVA including potential relevant variables mediating the results?

The discussion could include more rationale about previous studies, and the impact of the results found.

Reviewer 4 Report

Background:

-It is not rationalized well the focus on young people. Is there evidence or some precedent to suggest that young people should be affected more than other ages? Or if not, is there a reason why this study focuses on young people in particular? There are many articles written on the effects of COVID-19 across all populations and it is unclear what demographic factors confer the greater risk. 

Methods:

  • One of the central hypotheses about the difference between Turkey and Austria is the difference in lockdown measures. The questionnaires do not directly address the the participant's subjective experiences of the lockdown on their mental well being. Given the many, many mechanisms for increased psychological distress listed in the background, the questions do not clearly address the hypotheses of these mechanisms. 
  • There is a shortcoming of statistical analyses not controlling for some demographic measures that may be confounding the effects. For example, economic, occupational, and financial causes were listed in the background - the analyses should attempt to control for this. At the least it would help to have a results table that characterizes the averages for the sample (e.g. mean % employed, income, etc.)
  • There is no controlling for baseline psychological health and determinates of psychological health (employment etc.) prior to covid-19. There is no controlling for transient effects of stress that have no substantial long term effects on health or clinically significant effects.

Results:

  • need to define here or in the methods what "mental health status means". Mental health implies higher means greater health, however I wonder if the authors mean that higher status means worse mental health? This metric needs to be better defined

Discussion:

  • The conclusion that younger people in Turkey struggled more due to more stringent lockdowns is not supported by the fact that they were less likely to think the lockdown measures were "exaggerated" 
  • Significant differences in outcomes need to be put into context of clinically significant effects or not.. some of the mean differences were 1 point or less, which calls into question whether these differences are substantially relevant
  • Paragraph starting at line 290 - many of these findings are not properly discussed. It appears to lead to a confusing result that Austrians think about COVID-19 more and see it more dangerous yet have less psychological effects. This discrepancy needs to be explored.

Conclusions:

  • These conclusions are problematic. There is not sufficient justification that the "lockdown" is the active mechanism here for the purported effects. This suggestion is potentially harmful as well - lockdown measures may have significant public health benefits of reducing death, transmission, hospital overload, etc. The authors should responsibly balance these considerations as public health restrictions are necessary and while they cause stress may be worth it in the end. The clinical significance of these findings is not apparent - to suggest that the differences on a scale is equivalent or comparable to the death, morbidity, and healthcare strain of widespread transmission of covid-19 is a reckless conclusion
  • The alternative conclusion here is simply that covid-19 is a stressful situation and there is nothing that we can do about it - lockdown or no lockdown. Given that there is no controlled analysis of lockdown vs. no lockdown, these conclusions cannot focus making comments about the lockdown.
  • There needs to be a limitations section. There are many to this study and they need to be outlined. Importantly, there is no pre-post measure, to assess how transient these psychological effects may be. People, in general, are resilient; the majority recover from adversity. 
  • The other limitation is that there is no comparison to baseline prior to covid-19. For the many reasons already listed in the article, there are many reasons for worse psychological health at baseline in migrants even before covid-19 - -the pandemic may not even have made a change from baseline
  • there are many criticisms of "authoritarian" measures for lockdown that have shown evidence of benefit. however there has been some suggestion that strict lockdowns have been helpful however "draconian" - for example in China and other east asian countries

Round 2

Reviewer 2 Report

The article has improved a lot with the modifications made. Just one detail, lines 299 and 300 should not be separated.

Reviewer 4 Report

Thank you - the authors have adequately addressed my comments